# An Edge Computing System with AMD Xilinx FPGA AI Customer Platform for Advanced Driver Assistance System

**DOI:** 10.3390/s24103098

**Published:** 2024-05-13

**Authors:** Tsun-Kuang Chi, Tsung-Yi Chen, Yu-Chen Lin, Ting-Lan Lin, Jun-Ting Zhang, Cheng-Lin Lu, Shih-Lun Chen, Kuo-Chen Li, Patricia Angela R. Abu

**Affiliations:** 1Department of Electronic Engineering, Chung Yuan Christian University, Taoyuan City 320314, Taiwan; balance578942@gmail.com (T.-K.C.); panda19971209@gmail.com (J.-T.Z.); ine41748@gmail.com (C.-L.L.); 2Department of Electronic Engineering, Feng Chia University, Taichung City 40724, Taiwan; 3Department of Automatic Control Engineering, Feng Chia University, Taichung City 40724, Taiwan; yuchlin@fcu.edu.tw; 4Department of Electronic Engineering, National Taipei University of Technology, Taipei 10608, Taiwan; dtxion@gmail.com; 5Department of Information Management, Chung Yuan Christian University, Taoyuan City 320317, Taiwan; kuochen@cycu.edu.tw; 6Department of Information Systems and Computer Science, Ateneo de Manila University, Quezon City 1108, Philippines; pabu@ateneo.edu

**Keywords:** FPGA, edge computing system, advanced driver-assistance systems, deep learning processing unit

## Abstract

The convergence of edge computing systems with Field-Programmable Gate Array (FPGA) technology has shown considerable promise in enhancing real-time applications across various domains. This paper presents an innovative edge computing system design specifically tailored for pavement defect detection within the Advanced Driver-Assistance Systems (ADASs) domain. The system seamlessly integrates the AMD Xilinx AI platform into a customized circuit configuration, capitalizing on its capabilities. Utilizing cameras as input sensors to capture road scenes, the system employs a Deep Learning Processing Unit (DPU) to execute the YOLOv3 model, enabling the identification of three distinct types of pavement defects with high accuracy and efficiency. Following defect detection, the system efficiently transmits detailed information about the type and location of detected defects via the Controller Area Network (CAN) interface. This integration of FPGA-based edge computing not only enhances the speed and accuracy of defect detection, but also facilitates real-time communication between the vehicle’s onboard controller and external systems. Moreover, the successful integration of the proposed system transforms ADAS into a sophisticated edge computing device, empowering the vehicle’s onboard controller to make informed decisions in real time. These decisions are aimed at enhancing the overall driving experience by improving safety and performance metrics. The synergy between edge computing and FPGA technology not only advances ADAS capabilities, but also paves the way for future innovations in automotive safety and assistance systems.

## 1. Introduction

As artificial intelligence advances, computer vision is emerging as an important area for future application. Computer vision is the ability of computers to interpret and understand visual data from the environment. This technology has diverse applications, ranging from facial recognition [1] and object detection [2] to self-driving vehicles. The widespread use of cameras has generated an immense number of photos and videos. This vast amount of data is filled with different types of information, making it difficult to use it in an organized way. However, using machine learning and deep learning techniques, computers can now make sense of all these images and videos in ways never before possible.

Moreover, there has been a gradual shift from centralized to distributed computing duo to bandwidth limitations and resource requirements in the AI applications. The edge computing devices that perform AI computations locally have been discussed by [3,4]. In the area of edge computing systems, the advancement of ADAS is closely tied to the integration of edge computing and computer vision. Some of the technical challenges in ADAS were analyzed and discussed in [5,6]. However, some of these challenges can be addressed by current technology with the development of artificial intelligence. In the work of [7], the correlation between machine learning methods and ADAS was discussed. The study examined the applicability of the methods in different scenarios and provided insights into future research directions. With the current advancements in electric vehicles, the application of ADAS will necessarily rely on artificial intelligence or edge computing devices.

The current literature extensively explores the implementation of artificial intelligence in Advanced Driver-Assistance Systems (ADASs). Research encompasses various AI technologies such as machine learning and deep learning, with the overarching goal of enhancing vehicle perception, decision making, and control to bolster driving safety. Su et al. [8] utilized NXP’s embedded system as a platform for deploying the YOLOv3-tiny model, successfully mitigating computational overhead via their proposed architecture integrating SqueezeNet and quantization techniques with the MIPI camera. Azevedo et al. [9] proposed a real-time object detection and tracking solution for autonomous driving, optimizing deep learning models for edge deployment and successfully implementing them on the NVIDIA Jetson AGX Xavier platform with a USB camera. The existing commercial embedded system platforms offer robust computing capabilities, but their flexibility in integrating peripheral hardware is constrained by reliance on standardized interfaces.

In [10], systematic reviews were provided to analyze the implementation of ADAS. The FPGA could optimize the power consumption of the system and reduce reliance on the existing interface through hardware flexibility; however, the performance of the deep learning model is constrained by the limitations of hardware realization using high-level synthesis (HLS). Tatar et al. [11] presented a novel solution by implementing multi-task applications on FPGA using the DPU introduced by AMD Xilinx. The DPU specializes in model computations, effectively enhancing the performance of model operations on FPGA. Considering the resource constraints of edge computing devices, systems require a certain optimization capability. Currently, commercial embedded systems utilize existing hardware platforms to be edge computing devices. However, FPGA offers the flexibility to adjust hardware platforms and effectively plan software and hardware resources through heterogeneous architectures to achieve optimal resource utilization. After improving model performance, FPGA is a suitable choice for edge computing devices that require a balance of resources and performance. In the work of [12,13,14,15], researchers have proposed various ADAS applications effectively integrating FPGA platforms to achieve machine learning functions such as object detection. These studies have conducted extensive experimentation and provided valuable insights into model deployment and optimization. However, since the literature usually focuses on model accuracy and relies on off-the-shelf interfaces to communicate with external devices, the flexibility of FPGAs is not exploited.

This study utilizes the AMD Xilinx ZCU104 platform to enhance model computation performance through the DPU architecture, and integrates peripheral devices in a hardware-centric manner within ADAS applications. Machine learning models will be embedded into a custom circuit environment and coordinated with the operating system using PetaLinux. This approach optimizes the hardware platform based on resource constraints, thereby improving the overall performance of edge computing devices.

## 2. Methodology

The method of this study is the use of AMD Xilinx FPGA customer platform to implement pavement defect detection application. The system contains a vehicle camera, AMD Xilinx FPGA, and CAN signal transformer. The vehicle camera captures pavement information as a sensor. The system diagram for this study is shown in Figure 1. The raw image data will transition to the FPGA through signal lines. As the system’s central hub, the FPGA processes the I/O signals of peripheral devices, and, in conjunction with an AI computing unit, enables the detection of pavement defects. Then, the system transmits defect information to the backend device. To deploy the AI models, the FPGA must internally incorporate a processing system to coordinate various circuits and orchestrate the Deep Learning Processing Unit (DPU) for AI computations [16,17,18].

In the AMD Xilinx AI computing platform, it is critical to implement the PetaLinux operating system to invoke the DPU to perform model computations and retrieve the results. If there is a requirement for the operating system to invoke custom circuits, it involves writing a kernel and mounting it during the execution of PetaLinux. On the other hand, the AI model also needs to transform deployment on the DPU. In this study, the trained model will be quantized and compiled into specific files that the DPU can execute. These files are then invoked and executed through API calls. In ADAS, signals are primarily transmitted in the form of CAN. Consequently, it is necessary to convert detection results into CAN signals. This study utilized a boost model to elevate the FPGA’s output signal. Thus, Microchip’s MCU was employed for UART-to-CAN conversion. This method successfully transmits the detection results in the form of CAN signals.

Given the diverse functionalities requiring implementation on the FPGA, it is essential to utilize distinct development tools concurrently for the realization of circuit design and the deployment of AI models. In this study, using AMD Xilinx as an example, Vivado was utilized to design the circuitry for I/O signal processing, video signal processing, data communication, and to integrate the processing system with other circuits. In the AMD Xilinx Zynq architecture, processors are integrated with an FPGA, denoted as a processing system (PS) and Programmable Logic (PL). Communication between them is accomplished through the AXI interface, facilitating diverse applications. Through the Deep Neural Network Development Kit (DNNDK), the trained AI models can undergo pruning and optimization to fine-tune them for better compatibility with the FPGA platform. In preparation for deployment on the DPU, the pre-trained floating-point models will undergo quantization to become INT8 models. Subsequently, they will be compiled into an ELF object file for use as the DPU kernel in PetaLinux. PetaLinux is AMD Xilinx’s Linux-based operating system, enabling users to customize and deploy it on the PS. In PetaLinux, it is crucial to establish connections for all peripheral hardware and system components. This includes loading kernels and system packages, setting up interrupt signals, and so on. Finally, importing hardware files, kernels, and models results in the creation of a comprehensive operating system. The operating system is then enabled to run on the FPGA through boot configuration. With the Display Port (DP) kernel, the monitor displays the detection results on the camera scene and shows the progress of the data transfer. The details will be described in this section.

### 2.1. Customer Circuit Design

In this study, the hardware design in the proposed system can be categorized into three main functions. The first is the image source and image processing module, the second is the AI computing module, and the third is the detection results output module. The image source is captured by a commercial vehicle camera with a resolution of 1080p in the RGB888 format at a frame rate of 30 Hz. To mitigate the impact of interference in high-speed signal transmission, a module containing signal synchronization and filtering is implemented to serve as the image signal receiver in the FPGA. The FPGA circuit architecture for this study is illustrated in Figure 2. Employing an asynchronous FIFO as a buffer handles timing variations between camera and FPGA. This method enables seamless synchronization without the need for a shared clock, providing a reliable solution for overseeing image signal reception across diverse clock domains. By reducing high-frequency interference, the third-order digital filter ensures a clearer and more stable image transmission. This is vital in image processing, where maintaining signal integrity is crucial for optimal performance and clear visuals in AI computation. Due to the DPU’s requirements for AI model computation regarding image resolution, this study utilized a hardware resize module to adjust the camera resolution from 1080p to 416 × 16. This approach effectively improves efficiency in image transmission within the FPGA system. Another effective solution for image system is the utilization of Video Direct Memory Access (VDMA) [19,20,21]. Its primary function involves implementing DMA technology to efficiently move video data in video processing applications. VDMA features include frame buffers and video data stream. To implement VDMA in FPGA, collaboration between software and hardware is required. This section will focus on explaining the hardware aspect.

In FPGA, all image processing transfer interfaces utilized the Advanced eXtensible Interface (AXI), including VMDA, DPU, and the PS. Consequently, upon the FPGA’s acquisition of image data, the image data format undergoes transformation to the AXI format using the AXI transformation IP. The image data in the AXI format can be fed into the VDMA module by configuring parameters such as line buffer depth, stream data width, and enabling read/write channels based on the format of the image source. To execute AI model computations with the DPU, it is essential to utilize a PS equipped with an operating system to collaborate with DPU. Therefore, any task requiring collaboration with the DPU must be achieved through the PS. As the upstream provider of images to the DPU, VDMA needs to establish a connection with the PS and register within the operating system. By specifying the location of image access, the upper-level program can retrieve image data through function calls and dispatch it to the DPU for computation. Executable within the PL, the DPU necessitates instructions and accessible memory for input, temporary, and output data. AMD Xilinx provides a comprehensive API solution, allowing users to activate and control the DPU through the libraries. In terms of hardware, the DPU can be configured as required and connected directly to the PS. The system also provides appropriate clock signals for the DPU.

In this study, the integration of the camera, its corresponding driver circuit, the PS, and the DPU has been successfully achieved. Through meticulously designed hardware configurations, effective control and data transfer capabilities for the camera were realized. Furthermore, the integration of the PS and DPU within the framework of this study provides a synergistic hardware platform for image processing and deep learning tasks. This hardware circuit, combining several key components, offers high customizability and superior performance for the system under investigation. This system provides a hardware platform for implementing real-time image processing and deep learning tasks.

### 2.2. AI Model Platform with DNNDK and PetaLinux

In the context of an AI custom platform, DNNDK plays a crucial role in facilitating the deployment and optimization of AI models on specialized hardware, such as the DPU. It enables the compilation of models into executable formats compatible with the DPU architecture, ensuring the efficient and optimized execution of AI computations. Intrinsic to the DPU are processing elements (PEs) composed of DSPs, fetchers, and on-chip memory components. The AI models are compiled into instructions and stored in off-chip memory. These instructions are then fetched by the DPU’s fetcher via the AXI interface, and subsequently processed by the PEs. The system hardware architecture’s DPU is depicted as shown in Figure 3.

The DNNDK serves as the tools to transform AI models into DPU instructions. It comprises all tools essential for compiling and optimizing AI models, primarily categorized into three main functionalities: compression, compilation, and runtime. The tool for network compression is AMD Xilinx’s DECENT (Deep Compression Tool). DECENT addresses inference challenges in edge application by employing pruning and quantization, and reduces computational complexity while preserving accuracy. By converting 32-bit floating-point weights to 8-bit integers (INT8), DECENT enhances memory efficiency and inference speed. Its quantization calibration process requires minimal retraining and is complemented by quantize finetuning for improved accuracy. This approach significantly accelerates inference processes, making neural network models more suitable for deployment in edge computing environments. AMD Xilinx’s DNNC (Deep Neural Network Compiler) simplifies the deployment of deep learning models on AMD Xilinx platforms. It comprises a parser, optimizer, and code generator. The parser supports models from Caffe and TensorFlow, generating an intermediate representation (IR) for compatibility. The optimizer enhances model performance by optimizing computation graphs and reducing complexity. Finally, the code generator translates the optimized IR into efficient instructions for DPU, enabling models to run on the DPU. Deep learning applications designed for the DPU are heterogeneous in nature, comprising code executed on a host CPU, such as ARM, and code executed on the DPU. The compilation process for these DPU-accelerated applications involves several stages. Firstly, CPU code, typically written in C/C++, is compiled by compilers like GCC or LLVM. Concurrently, the DNNC compiles computationally intensive neural networks into DPU binary code for acceleration. In the final step, the CPU and DPU code are linked together by a linker, such as GCC, to generate a single hybrid binary executable. This executable contains all the necessary components for heterogeneous execution on both CPU and DPU. The DPU runtime, facilitated by DNNDK, plays a vital role in the execution of deep learning applications on the DPU. These applications are compiled and linked into a hybrid binary executable, which resembles standard applications. Beneath the surface, a combination of the standard Linux loader and the DPU loader manages the loading and execution process of these hybrid applications. The DPU runtime environment consists of essential components such as the DPU Loader, DPU profiler, DPU runtime library, and DPU driver. The DPU Loader, in particular, facilitates the transfer of DPU kernels from the hybrid ELF executable into memory, dynamically relocating the DPU code as necessary for efficient execution.

PetaLinux is a comprehensive Linux operating system developed by AMD Xilinx for customizable deployment on the PS of FPGAs. It enables users to tailor and deploy Linux-based solutions specifically for FPGA-based platforms. PetaLinux facilitates the establishment of connections for peripheral hardware and system components within the FPGA environment. This includes tasks such as loading kernel configurations, configuring system packages, and setting up interrupt signals. PetaLinux facilitates the efficient customization and optimization of FPGA-based systems, meeting tailored requirements for specific application needs while retaining the flexibility and power of Linux-based operating environments. The development process in PetaLinux involves several key steps. Initially, custom circuit synthesis and export are conducted using Vivado, encompassing hardware architecture design and definition, ultimately resulting in a bitstream file. Subsequently, a customized project is established within PetaLinux, setting up the project structure and configurations. Following this, configurations are performed based on users’ applications. PetaLinux’s configuration interface facilitates the adjustment of various system parameters, including filesystem, kernel, and DPU settings. Upon completing the configuration, the operating system takes place. PetaLinux automatically constructs and synthesizes the system according to the configuration, generating an image file containing the root filesystem and kernel. Finally, the boot image is created and employed to initiate the processing system on the FPGA via an SD card.

DNNDK and PetaLinux are crucial for implementing deep learning models on the AMD Xilinx FPGA platform. DNNDK handles the processing of deep learning models and their deployment onto FPGA for efficient inference processing. PetaLinux serves as the core development platform for FPGA, providing a Linux-based operating system that seamlessly integrates the FPGA’s PS and PL. In this study, YOLOv3 serves as the test model, transformed using DNNDK and deployed onto the DPU through DNNDK’s quantization, enabling inference processing for image recognition. Additionally, hardware design information is imported into PetaLinux, generating an embedded Linux system that effectively integrates FPGA resources and provides the necessary operating environments. The combination of these two tools allows for real-time image recognition and processing, fulfilling the requirements for edge computing functionality in FPGA.

### 2.3. CAN Signal Conversion

The automotive networks used in commercial vehicles can be classified based on protocol characteristics and transmission rates into Local Interconnect Network (LIN), Controller Area Network (CAN), Media Oriented System of Transport (MOST), and FlexRay. CAN, as a real-time, distributed communication protocol, shares information on the bus through message frame exchanges, offering high real-time performance. It is widely utilized in the automotive domain. In ADAS applications, unnecessary image transmission lead to inefficiency and resource wastage. Therefore, it is imperative to organize computational results into automotive control interfaces. This enables Vehicle Control Units (VCUs) to make control decisions based on computational outcomes. By doing so, the system optimizes resource utilization and enhances overall efficiency, aligning with the requirements of ADAS applications.

In FPGA development, efficient communication interfaces are vital for transmitting signals. One common interface used is UART, which facilitates serial communication between devices. To integrate UART functionality into the FPGA hardware design, a UART IP core is instantiated and connected to the PS, while also defining the GPIO pin. This connection allows the FPGA to communicate with external devices or systems using the UART protocol. Once the UART hardware is designed and implemented, it must be configured within PetaLinux. This involves updating the device tree to register the UART hardware, specifying interrupt signals, and defining memory locations. In addition, during the boot process, the UART kernel needs to be loaded to enable UART functionality in the Linux environment. With the UART hardware configured and the kernel model loaded, the CPU program can now utilize the UART interface to transmit data via GPIO. This enables seamless communication between the FPGA-based system and external devices, facilitating various application such as data logging, and debugging or interfacing with other embedded systems. Through proper integration and configuration, UART enhances the versatility and connectivity of FPGA-based systems in diverse applications.

In this study, a Microchip MCU is utilized for the conversion of UART to CAN signals, a crucial component in enabling communication between different automotive systems. Microchip’s MPLAB X IDE is employed for the development of embedded system software, providing a comprehensive platform for programming and testing. The MCU program is designed to efficiently organize and transmit recognition results obtained from the FPGA as CAN messages. Acting as a bridge between the FPGA-based system and the CAN network, the MCU facilitates seamless communication and integration of the FPGA’s image recognition capabilities into automotive back-end systems. This integration enhances the overall functionality and performance of the automotive system, enabling efficient data exchange and decision-making processes. Through this methodology, this study aims to demonstrate the feasibility and effectiveness of incorporating FPGA-based image recognition into automotive applications, paving the way for enhanced safety and efficiency in vehicle systems.

## 3. Device Implementation

In the context of ADAS, the integration of FPGA-based image recognition coupled with AI computation represents a significant advancement in automotive technology. This experimental study focuses on the implementation and validation of such a system, aiming to enhance vehicle safety and performance. The experimental setup involves the deployment of FPGA hardware equipped with image recognition capabilities, powered by AI algorithms. Utilizing the computational power of FPGA, real-time image processing tasks such as object detection are performed efficiently. The FPGA hardware is programmed to analyze the input video stream from onboard cameras, extracting relevant information for driving decisions. To validate the system, this research utilizes a DisplayPort (DP) to output the screen for observing the display’s correctness. The FPGA’s recognition results are marked on the screen through a CPU program to confirm the accuracy of recognition. A CAN signal monitoring device, PCAN, is used to verify the signal transmission.

To validate the system, this research employs a multi-faceted approach. Firstly, the DisplayPort (DP) is utilized to output the screen, allowing for the visual inspection of the displayed content to ensure its correctness and integrity. Additionally, a CPU program is utilized to overlay the recognition results generated by the DPU onto the screen. This enables us to visually confirm the recognition system. Furthermore, signal transmission verification is conducted using a monitoring device known as PCAN. This tool is employed to monitor the transmission of CAN signals, ensuring that they are properly transmitted and received by the intended recipients. By using the PCAN, the reliability and robustness of the communication channel can be validated, and it can be confirmed that important data are successfully transferred between the different components of the ADAS. Overall, this comprehensive validation approach ensures the accuracy of the FPGA-based image recognition system used in the ADAS application. The process and details are described below.

### 3.1. System Hardware Architecture

For the realization of real-time edge computing in ADAS, the AMD Xilinx platform is utilized in this study. Using the FPGA platform enables a swift and flexible execution of circuit design, simulation, and verification tasks. Moreover, the platform offers ample computing resources for edge computing, enhancing the system’s performance. The Xilinx FPGA platform was selected as the implementation platform in ADAS for rapid and accurate development, evaluation, and deployment. The AMD Xilinx ZCU104 FPGA was selected as the central component of this system. The system requirements include image input, deep learning model computation, and automotive interface signal output. The image input section utilizes commercial automotive cameras as image sources, which are connected to the FPGA via GPIO. This configuration enables the FPGA-based system to receive real-time video feeds from the vehicle’s environment, facilitating various vision-based applications. The image source format is 1080p RGB888 at 30 fps. This study focuses on the layout diagram of automotive cameras and FPGA circuits, as shown in Figure 4. Due to impedance issues in the transmission line, additional synchronization and filtering are required on the FPGA as the receiver. After image processing and recognition tasks on the FPGA, the recognition results are transmitted via the UART interface. Then, the results are sent to the back-end device for UART-to-CAN signal conversion.

The UART-to-CAN conversion was implemented using the Microchip’s PIC18F46K80-I/PT MCU, a versatile microcontroller known for its reliability and flexibility in embedded systems. This MCU enables communication between devices operating on UART and CAN protocols, bridging the gap between different communication standards commonly found in automotive applications. The integration of this microcontroller into the ADAS enables the efficient conversion of UART signals from the FPGA-based vision system into CAN messages, which are then transmitted to other automotive components via the CAN interface. This enables real-time data exchange and coordination between different modules, enhancing the overall performance and functionality of the ADAS. However, a voltage level mismatch occurs because the FPGA GPIO output pins operate at a maximum voltage of 3.3 V, while the input voltage requirement for the MCU module is 5 V. To bridge this gap, a voltage boost module is required to increase the voltage level and meet the MCU input requirements. In this study, a commonly used UART-to-RS232 voltage boost module was used for this purpose. The reliability and effectiveness of the voltage boost module is confirmed by voltage measurements using a voltmeter to ensure that the voltage level is raised to the required 5 V. This verification process enables communication and compatibility between the FPGA and MCU modules.

### 3.2. FPGA Circuit Architecture

In the implementation of an ADAS system, automotive cameras serve as the primary image source. In addition to the hardware setup, attention must be paid to signal processing to ensure stable reception, especially for high-resolution and high-speed signals. Jin et al. [22] demonstrated the use of two image input sources, digital video and camera, using a camera interface layer to process these signals. Tang et al. [23,24] introduced alternative image sources from peripheral devices. In particular, digital video can output fully parallel color formats, while camera modules typically transmit color signals sequentially. Signal reception is handled by the frame grabber module within the Camera Link interface, which converts the signals into image formats compatible with other devices. To ensure the stability of the image signal, a synchronization module and filtering module have been integrated into the frame receiver circuit. This module used a first-in-first-out (FIFO) buffer to handle temporal problems between signals. In addition, a three-stage digital filter circuit is implemented to eliminate noise and prevent it from affecting the detection accuracy. The input pins of frame receiver module include 1 bit each for pclk (pixel clock), hsync (horizontal synchronization), vsync (vertical synchronization), href (horizontal reference), and 8 bits for data. These input signals are routed through the pclk input to the FIFO where they are synchronized to the local clock. This synchronization acts as a trigger condition for the FIFO output, ensuring the proper operation of subsequent module.

The AMD Xilinx FPGA ZCU104, featuring Zynq UltraScale+ MPSoC, integrates ARM processors and FPGA logic, making it ideal for ADAS applications. Before forwarding image data for deep learning processing, storage and access methods are carefully considered for optimal real-time performance. The communication interface used the AXI-Bus to ensure compliance with the AXI interface standard. To facilitate the execution of deep learning models, image data are resized to a specific resolution for the DPU. In FPGA, image data is stored in DDR memory for access by the DPU. However, relying on the processor for each data access operation significantly increases the system execution time. To address this issue, AMD Xilinx provides a VDMA solution that can be directly activated via an API and automatically operated based on memory address, reducing communication time when accessing large amounts of image data. To deploy the deep learning model, this study leverages the DPU architecture offered by AMD Xilinx. The DPU IP from AMD Xilinx is integrated into the system, establishing a connection with the ARM core through this architecture. The hardware architecture adheres to the official DPU targeted reference design (DPUTRD), as shown in Figure 3.

To efficiently communicate image recognition results within the ADAS application, the AMD Xilinx UART lite IP core was used in this research. Integrated into the FPGA design, this IP core facilitates communication using the UART protocol. The IP core bridges the FPGA’s internal processing unit with external systems, facilitating the transmission of detection results. During the implementation, it is essential to configure and activate the UART driver in PetaLinux, as well as to ensure that the GPIO pins of the FPGA are correctly connected to the TX/RX pins of the UART interface. The FPGA hardware architecture is a versatile framework that integrates multiple modules to support complex embedded system functionalities. By combining modules such as grame grabber, resize, VDMA, processing system, DPU, and UART Lite, FPGA-based systems efficiently process image data, perform deep learning inference, and communicate with external devices. This integration successfully realized FPGA’s edge computing capabilities, effectively applied in ADAS applications.

### 3.3. YOLOv3 with DNNDK

Device implementation in the context of deep learning involves the transformation of trained models into a format suitable for deployment on target hardware, such as DPU. The PetaLinux build process can be referenced in Figure 5. The AMD Xilinx platform provides DNNDK to meet this application’s needs, with a particular focus on real-time inference, which is critical for ADAS. The DNNDK toolset supports two framework modes: Caffe and TensorFlow. This study used the YOLOv3 model, which is based on the Caffe framework. Thus, all procedures described in this study follow the Caffe mode. The workflow begins with DECENT, which modifies the floating-point model by quantizing weights, biases, and activations to reduce computational complexity and memory requirements. DECENT also provides calibration functionality that iteratively refines quantization to improve accuracy. By providing a list of datasets, iterative calibration fine-tunes quantization parameters, improving the accuracy of model inference. After the DECENT operation, specific adjustments are made to the model files, such as the prototxt file and the Caffe model file, to optimize hardware performance. DNNC facilitates the conversion of quantized deep learning models into deployable ELF files for DPU deployment.

In the field of ADAS application, the development of an advanced perception system is crucial for intelligent driving capabilities. Enhancing perception and detection capabilities through various technologies is a common goal. This study focuses on the use of image sensing to detect pavement defects such as puddles, potholes, and bumps. The pavement detection model was trained using an image-based artificial intelligence algorithm and was deployed on an embedded system for integration into ADAS application. The selection of the model was based on its compatibility with DPU and its ability to perform object recognition. In this study, YOLOv3 was selected as the deep learning model for verification. Due to the pavement defects are major contributors to damage in vehicle chassis and suspension systems, posing significant risks to the safety of both drivers and pedestrians [14]. This model was trained using the TPDID (Taiwan Pavement Defect Image Dataset), which consists of real pavement defect data from different locations in Taiwan. The dataset also includes various lighting conditions, including night, low light, and rainy, to enhance its comprehensiveness and optimize the model’s recognition capabilities. TPDID includes seven defects for pavement detection, namely, deceleration ramp, water pothole, pothole, manhole cover, puddle, expansion joint, and road damage. Figure 6 illustrates three of these defects within TPDID.

### 3.4. Patalinux Configuration

The PetaLinux synthesis process is used to equip an FPGA with a Linux system. This includes initializing a project, setting up the root file system, adjusting kernel settings, and configuring parameters relevant to the DPU. PetaLinux project files are then synthesized to generate BOOT.bin and image.ub files that facilitate the deployment of a Linux system on the FPGA. This approach ensures efficient management and customization of Linux systems tailored for FPGA platforms, thereby improving integration and performance in various applications. Firstly, the PetaLinux tools suite needs to be installed on the workstation. Then, a new PetaLinux project is created by running the corresponding command. Depending on the application requirements, additional packages are added to PetaLinux. This study provides packages specifically tailored for ADAS with DPU, including the DNNDK package and DPU package. In addition, OpenCV packages are included for the convenience of image processing using the OpenCV library, which facilitates CPU program handling of images. The corresponding packages are listed in Table 1.

Device tree registration is the process of describing hardware components and their configurations in a structured formant. This ensures that the Linux kernel understands the hardware present on a system. Each device is assigned a compatible string that matches a corresponding driver in the Linux kernel. In addition, device-specific properties such as interrupts, memory addresses, and bus connections are specified. To register the DPU, a new node for the DPU was added to the device tree structure. This node contains declarations for the base address location, number of DPU cores, and interrupt characteristics based on the hardware design. Similarly, to control the UART Lite IP, another node is required in the device tree, specifying relevant parameters according to the hardware characteristics. This process ensures that the Linux kernel recognizes and properly configures both the DPU and the UART Lite IP, enabling seamless integration and the use of these hardware components within the operation system. Finally, after configuring the PetaLinux project and device tree, the project is built using the specified commands to generate the system image files, BOOT.bin and image.ub. Since the ZCU104 board is equipped with an SD card slot, the SD card must be partitioned as FAT32 using the Gparted software (version 1.6.0). Once the partitioning is complete, the generated system image files can be burned to the SD card. Through this process, the FPGA’s processing system can run with an operating system, allowing peripheral hardware such as the DPU and UART Lite to be controlled by the CPU program to meet application’s requirements.

### 3.5. CPU Program

In the CPU program, the execution flow begins with the initialization of necessary resources and data structures. The CPU data processing flowchart is illustrated in Figure 7. This includes setting up communication interfaces, loading deep learning models, and allocating memory. Once initialized, the program enters a loop or begins to execute functions asynchronously, depending on the design of the application. In this study, the image processing pipeline continuously reads and processes images from a camera. With this processing loop, various functions perform specific tasks. The CPU program consists of six functions: FrameReader, ImagePreprocess, RunModel, ResultProcess, OutputUART, and DisplayScreen. These functions are called and coordinated by the main program to retrieve images from the VDMA, process them using DPU, and transmit recognition results via the UART while displaying real-time content via the DP.

First in line is the FrameReader function, which extracts frames from the VDMA and appends them to a queue. These frames are environmental information captured by the camera and serve as input data for processing. Once the frames are obtained, the ImagePreprocess function refines the image data to ensure that it meets the requirements of YOLOv3. This pre-processing phase includes defining the size of the input images by the neural network and dequeuing frames from the FrameReader to prepare a sequence for input to the DPU. With the images appropriately pre-processed, RunModel executes the YOLOv3 algorithm, using the DPU’s processing power to perform real-time detection and localization within the captured images. After executing the YOLO algorithm, ResultProcess engages in post-processing for DPU output. It detects object positions and classes, annotates the original image with bounding boxes and labels, and computes class counts for output. OutputUART then converts the results into a UART-compatible data format, allowing the detection results to be transmitted over the UART interface. This facilitates communication with the UART-to-CAN module. Meanwhile, the DisplayScreen presents the processed frames in real time with bounding boxes and class labels via DP. This visual feedback provides a conducive debugging environment and ensures system operation. The final step is to execute the application, which includes tasks such as initializing resources; loading the UART device and DPU; loading the YOLOv3 model; creating DPU tasks; spawning threads for video reading, YOLOv3 execution, and frame display; and ensuring proper resource release upon completion. In the FPGA processing system, the CPU program plays a critical role in the execution of the entire application. As the central control hub, it coordinates the various functions and components involved. This successfully realized an ADAS application with edge computing capabilities.

## 4. Experimental Results and Device Implementation

This study focuses on detecting road surface conditions by examining the training and testing results of YOLOv3 on multiple images depicting various road defects. Figure 8 demonstrates the process of displaying the original image for contextual understanding of the road condition, and presents the test result, which highlights the model’s effectiveness in accurately identifying and localizing road defects. These findings hold significance for enhancing road surface issue detection and autonomous driving safety, thereby contributing to the development of systems aimed at detecting and responding to road surface conditions in the context of autonomous driving.

When verifying an edge computing device for an ADAS application, it is important to ensure its functionality and reliability. Edge computing plays a critical role in ADAS because it is responsible for processing sensor data in real time, enabling quick decisions to improve driving safety. In the verification process, a monitor and PCAN with PC are essential tools. The monitor is connected to the DP connector on the ZCU104 to confirm the output results from the DPU. This visual inspection ensures the accuracy and correctness of the DPU’s output. Meanwhile, the PCAN is connected to the output pins of the MCU module. This setup allows for the confirmation of the information output, ensuring that the recognition results are properly transmitted as CAN signals. By leveraging these tools, the verification process validates the functionality and performance of the entire system, ensuring its reliability and effectiveness in real-world applications.

The verification process is divided into three stages. First, to confirm that the system boots correctly and enters the operating system interface, it is ensured that the DP function operates normally Second, VDMA verifies that camera data are sent to memory and retrieved via API, and the screen displays the video from the camera. Finally, it runs the main code to execute the entire system. When the system is operating normally, detected objects can be observed and marked on the video screen in real time. In addition, the terminal window shows the object data being transferred. The output UART signals are boosted in voltage by the booster module and converted to CAN signals by the MCU module. The PCAN is used in this study to verify the accuracy of the CAN signals by capturing and analyzing them. The dedicated software interface is used to verify that the CAN signals are captured accurately and contain correct recognition information. Due to the difficulty of reproducing road conditions and the challenges of on-site debugging, the verification environment simulates driving situations by playing driving videos on the screen. As shown in Figure 9, the verification environment consists of two screens: one displaying the operating system user interface and the other playing driving videos as the image source. A camera is placed in front of the video screen, and the captured image data are processed by the FPGA using deep learning computations. The result data are transmitted via the UART, and the voltage is boosted by the booster module. Then, the signal is converted into CAN signals by the MCU module and output to the PCAN to verify the correctness of the CAN signals. Once the DPU computation is complete, the results are stored in memory. The CPU program labels objects on the video frames captured by the camera based on their type and pixel addresses. Thus, it is possible to observe on the screen whether objects have been correctly recognized by the algorithm. As shown in Figure 10, the deceleration ramp in the frame is successfully identified by the DPU and accurately labeled at the corresponding pixel location.

Table 2 and Figure 11 provide the data format for CAN signals to facilitate the transmission and display of related information about detected objects. Each CAN ID corresponds to specific object details, including object category, distance, center coordinates (*X*-axis and *Y*-axis), and object length and width. CAN ID ‘0xE1’ represents the total number of detected objects, with unique CAN IDs assigned to each different object category. This information can be used to analyze and visualize the performance of the vision system in identifying objects in the environment. Thus, it helps to evaluate the accuracy and efficiency of the system. In this study, the Ethernet interface on the FPGA was used to enable remote control of the FPGA from a PC running PCAN-Viewer to capture PCAN-Viewer and operating system screens simultaneously. By referencing both screens simultaneously, it is easier to observe whether the recognition results match the contents of the CAN signals. As shown in Figure 12, the deceleration ramp is successfully recognized and accurately displayed in the PCAN-Viewer. The system architecture proposed in this study successfully realizes an FPGA-based edge computing device and applies it to the ADAS application.

In this study, the use of a 1xB4096 architecture DPU on the XCZU7EV platform for pavement defect recognition tasks involved 416 × 416 pixel images. The YOLOv3 model, which has a size of 240.59 MB and operates with INT8 precision, was effectively managed by the DPU operating at a frequency of 300 MHz, delivering a frame rate of 10–15 FPS. To optimize performance and offload computational tasks from the CPU, certain functionality was implemented on the PL side of the FPGA. The hardware-based solutions were implemented for image pre-processing and model inference output tasks, including resizing and UART communication. These solutions replaced traditional software-based methods, such as OpenCV for resizing and video display. By offloading computational tasks from the PS to the PL, this approach resulted in a significant improvement in overall system performance. In addition, custom modules were integrated into the circuit architecture with the DPU and a custom Linux-based operation system was successful with resizing and UART communication functionality. The resource usage data shown in Table 3 highlight the distribution of FPGA resources, with particular emphasis on the impact of resizing and UART modules. On closer inspection of the resizing and video display operations, it was found that improving the resizing process had little effect on the FPS improvement, but replacing the video display functions with the UART resulted in a significant improvement in system performance. This finding highlights the importance of strategically optimizing hardware functionality to achieve optimal system performance. The FPGA resource utilization for this study is presented in Table 4.

Despite the relatively low utilization of FPGA resources by the custom circuits comprising the image receiver, resize, and UART modules, their impact on overall system performance is significant. This highlights the effectiveness of using FPGA-based solutions for specific tasks and underlines the importance of hardware-level optimizations in improving system efficiency. In contrast to the previous literature focusing on model accuracy and quantization distortion, this study discusses methods for optimizing system performance at the hardware level within the heterogeneous computing architecture of the FPGA, integrating both PS and the PL.

## 5. Discussion

This study presents the design of an FPGA architecture that integrates a custom circuit with the DPU, effectively improving system performance. The image pre-processing module designed within this architecture is connected to the CPU via VDMA, facilitating image processing cooperation between the CPU and the DPU. The computation results are then transferred from the CPU to a UART module via the AXI interface, enabling the successful implementation of ADAS application. Experimental results showed that the custom circuit effectively improved system performance, particularly when the image display function was replaced by the UART module. The custom circuit, which used less than 1% of the FPGA’s resources, increased system performance by approximately 50%. It was observed that tasks with lower CPU computational demands showed minimal performance gains from hardware optimizations, suggesting that focusing on functions with higher computational demands yield greater benefits. In the design of the image pre-processing and UART modules, the successful integration of external devices via GPIO interfaces bypassed the platform’s hardware connector limitations and provided a solution for optimizing system performance. The flexibility of FPGA has been thoroughly leveraged in our study, yet maximizing its advantages poses challenges. In the optimization process, a primary concern is balancing performance with resource utilization and power efficiency. To address this, optimized algorithmic designs and high-level synthesis tools can be employed, alongside considerations of practical application costs. Integrating more advanced AI models into FPGA presents opportunities and challenges. Key issues include resource utilization, algorithmic complexity, harnessing parallel processing, enhancing memory bandwidth, power conservation, AI framework integration, and meeting real-time requirements. This approach is particularly valuable for maximizing resource utilization under the constraints of edge computing devices.

Despite these findings, the discussion of FPGA flexibility remains multifaceted, encompassing topics such as CPU scheduling and DPU acceleration architectures. These directions represent critical avenues for further research to fully exploit the potential of FPGA-based edge computing in deep learning applications. In conclusion, this study presents an FPGA circuit design incorporating a Deep Learning Processing Unit and a Linux-based kernel for efficient ADAS applications. Our research results provide valuable insights into hardware optimization strategies and their impact on system performance in the FPGA-based deep learning domain.

## 6. Conclusions

Deploying deep learning models on FPGA platforms expands their range of applications. Coupled with the integration of peripheral circuitry, FPGAs become efficient edge computing devices. This approach leverages the computational power and flexibility of the FPGA to execute complex real-time object detection and recognition algorithms. It is used to enhance the intelligence of ADAS application by applying pavement defects’ detection. The FPGA system benefits from accelerated inference capabilities and reduced latency, enabling faster and more efficient processing of image data. This integration facilitates the efficient deployment of deep learning models directly on edge computing devices. As the system complexity increases, FPGA and its comprehensive development tools also effectively reduce development time In this study, a vehicle camera was used to provide image sources to the FPGA. Inside the FPGA, an image pre-processing circuit was established that includes filtering, synchronization, resizing, and VDMA. The processing system with PetaLinux allows the coordination with peripheral circuits, especially in the delivery of image data from the VDMA to the DPU for computation. To better align the system with ADAS applications, this research also incorporates a UART signal module at the output of the FPGA to connect the signal conversion device. This allows the CAN signals to be transferred to the vehicle controller for overall vehicle control. The proposed approach has been effectively implemented on FPGA by integrating deep learning models, custom circuits, and an operating system. This enhancement significantly improves the performance of ADAS systems with pavement defect detection applications. In this study, custom circuit for image resizing and UART communication was implemented to offload CPU tasks related to image pre-processing and output. This strategy resulted in a significant improvement in system performance with only a minimal increase in resource utilization. Leveraging the heterogeneous computing architecture of highly flexible FPGAs, both hardware and software optimizations maximized the benefits of the FPGA, making it more suitable for resource-constrained edge computing devices.

This study explored hardware optimization techniques to improve system performance and showed that tasks with higher CPU resources requirements benefit from hardware optimization. The proposed architecture utilized GPIO interfaces to overcome existing platform limitations and successfully implemented a custom circuit with the DPU to improve the system’s speed. This research contributes to an optimized approach for FPGA utilization in deep learning applications. In this study, the YOLOv3 framework that was supported by the DPU was used to train a model by the TPDID dataset. In particular, it was easier to demonstrate the improved performance achieved by integrating custom circuit under the same DPU architecture as the official specifications. However, the high flexibility of FPGAs prompts further discussion on various aspects such as model quantization distortion, CPU acceleration methods, DPU scheduling optimization, or custom deep learning computation units. Overall system optimization requires consideration of multiple facets. Future research directions could explore advanced FPGA architectures for optimizing deep learning inference tasks, and investigate novel hardware acceleration methods tailored to specific application domains. In addition, further studies could focus on refining custom circuit designs to maximize FPGA resource utilization and improve system efficiency.

## Figures and Tables

**Figure 1 sensors-24-03098-f001:**
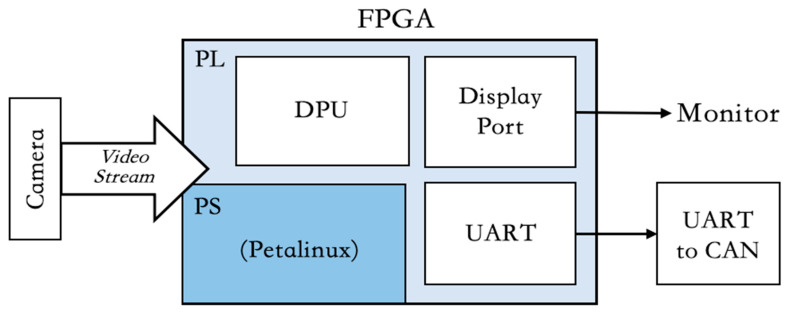
System Architecture Diagram.

**Figure 2 sensors-24-03098-f002:**
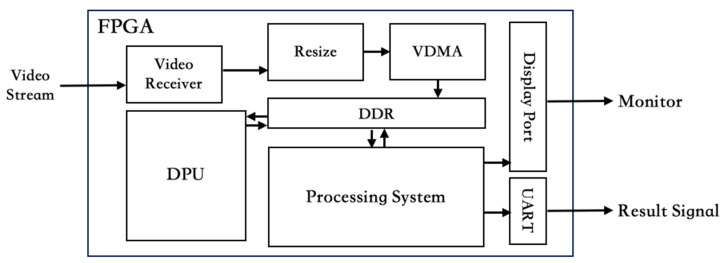
FPGA Circuit Architecture Diagram.

**Figure 3 sensors-24-03098-f003:**
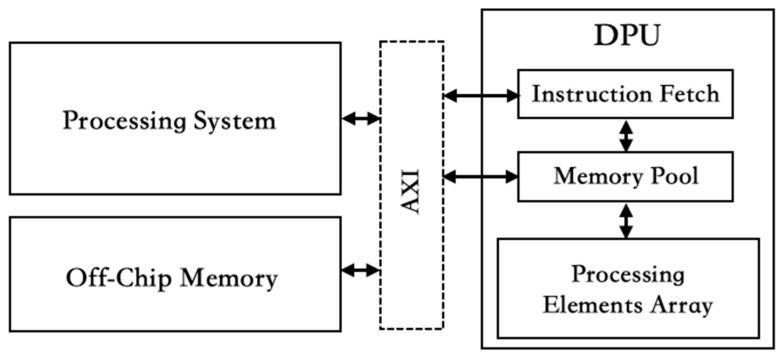
DPU with Processing System Hardware Architecture.

**Figure 4 sensors-24-03098-f004:**
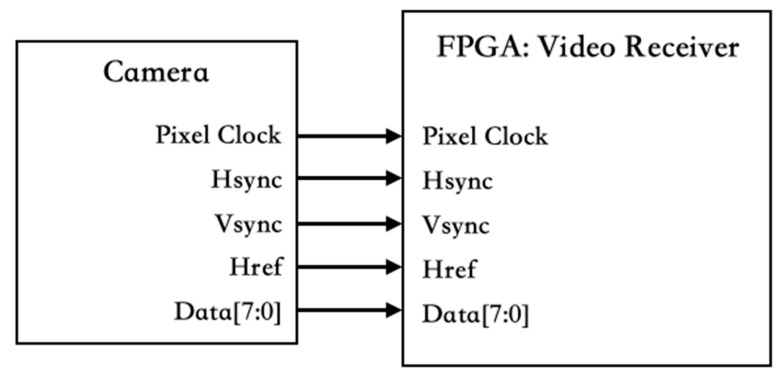
Camera and FPGA Pinout Diagram.

**Figure 5 sensors-24-03098-f005:**
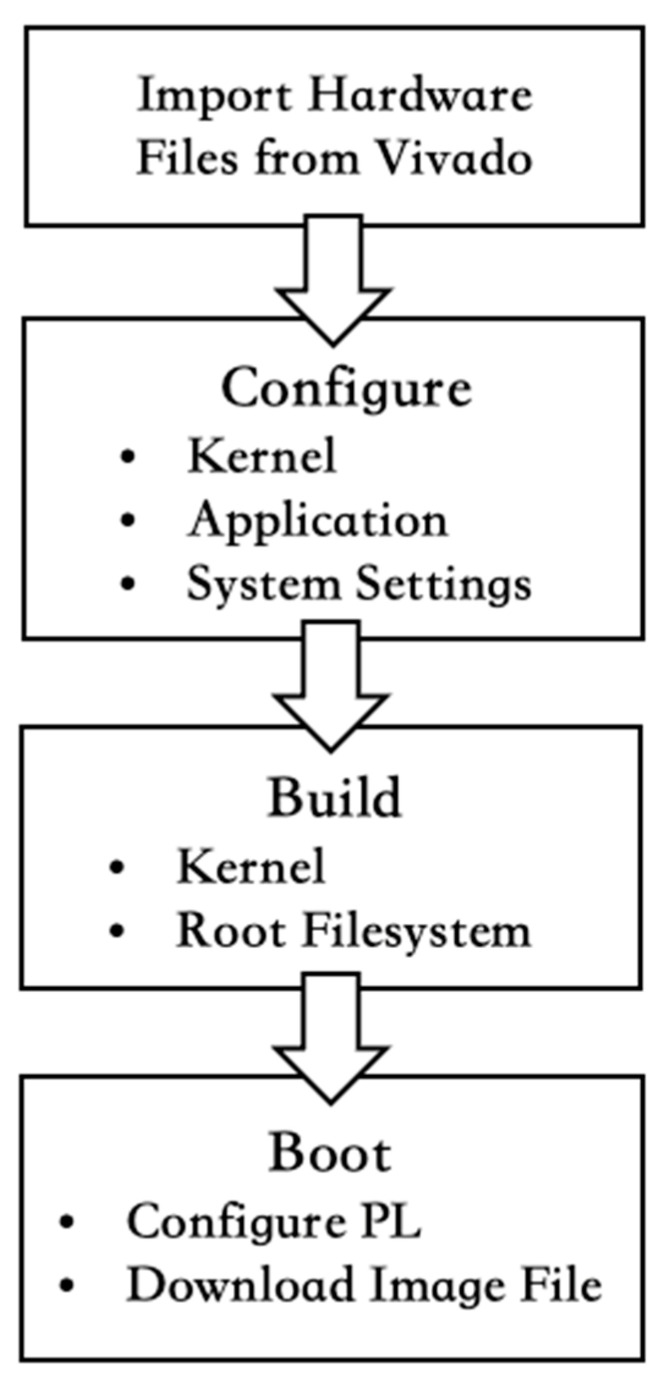
PetaLinux Building Process.

**Figure 6 sensors-24-03098-f006:**
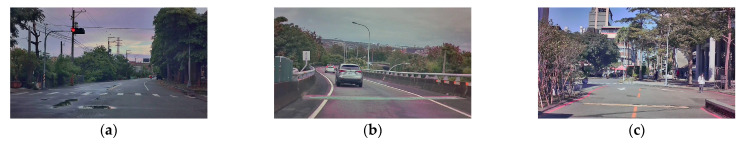
The Pavement Defects Sample in TPDID. (**a**) Puddle; (**b**) expansion joint; and (**c**) deceleration ramp.

**Figure 7 sensors-24-03098-f007:**
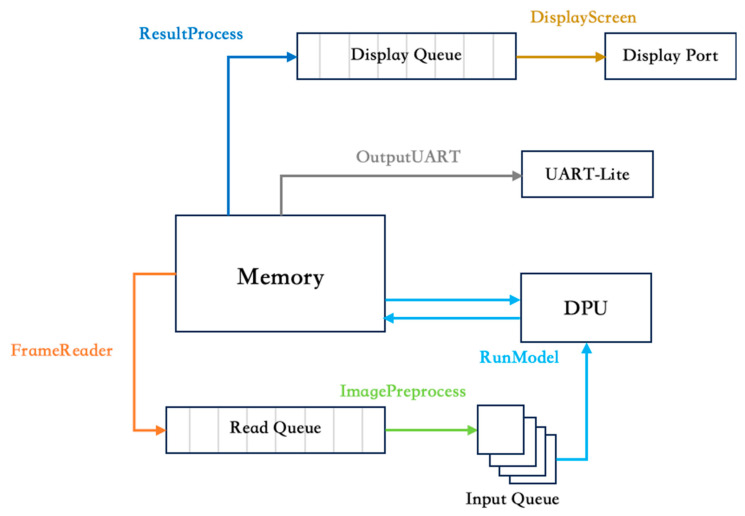
CPU Program Data Flow Diagram.

**Figure 8 sensors-24-03098-f008:**
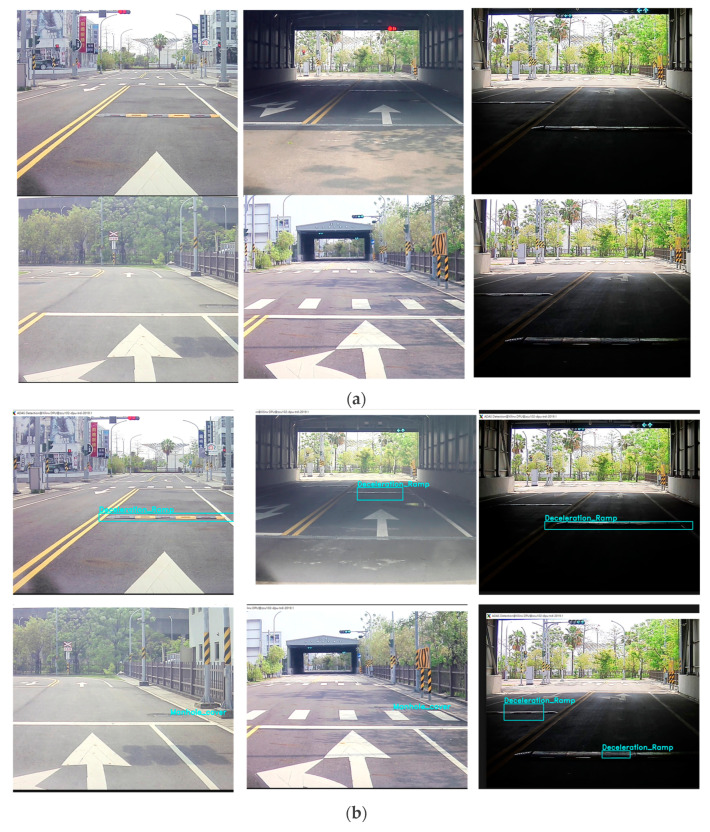
Training and testing results of YOLOv3 on multiple images in this study. (**a**) Original image and (**b**) test result.

**Figure 9 sensors-24-03098-f009:**
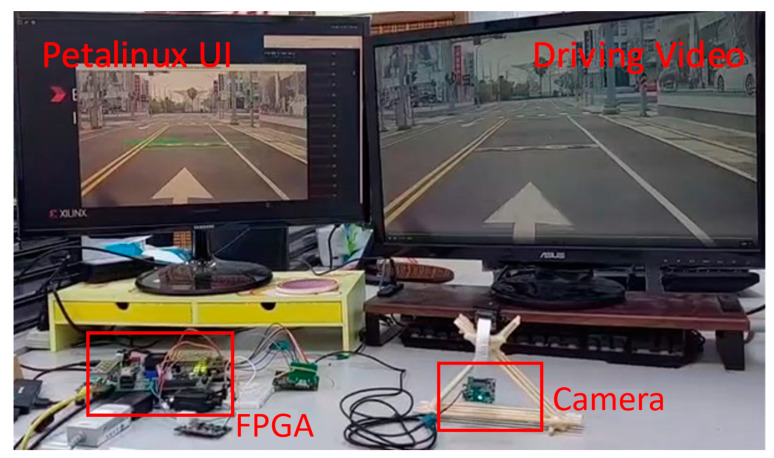
System Verification Environment.

**Figure 10 sensors-24-03098-f010:**
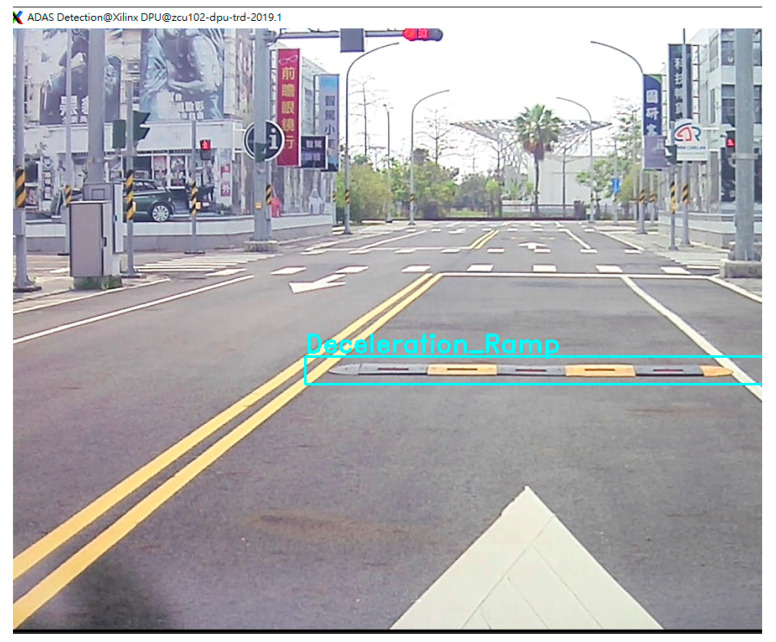
Deceleration ramp detection.

**Figure 11 sensors-24-03098-f011:**
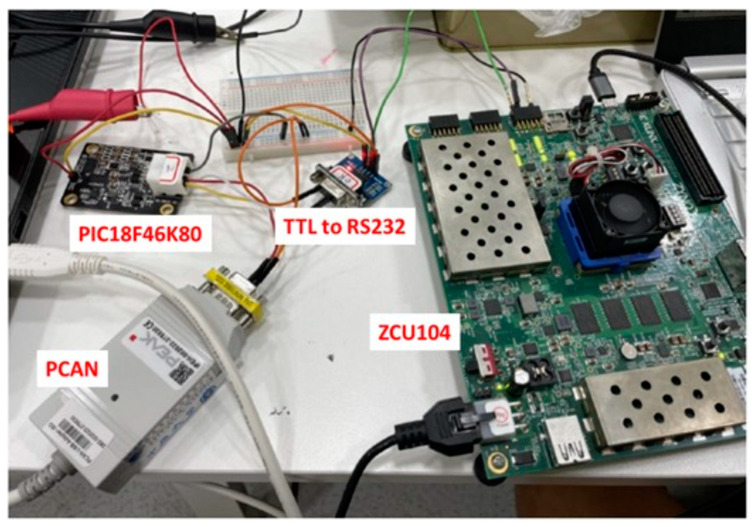
UART to CAN System.

**Figure 12 sensors-24-03098-f012:**
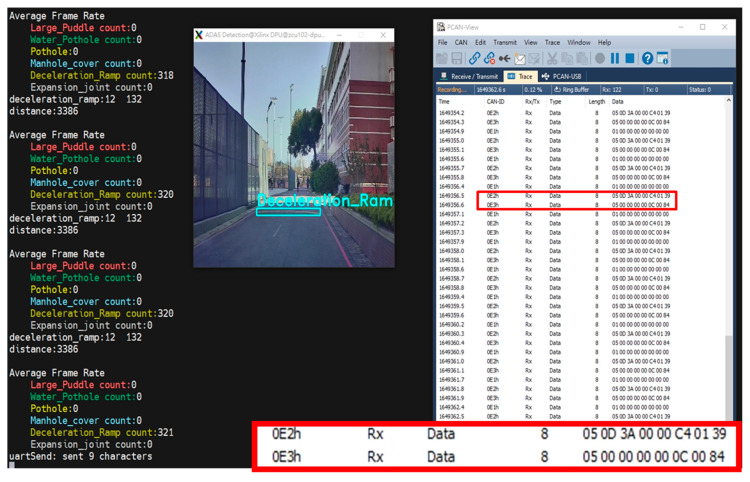
Recognition results and PCAN-viewer UI.

**Table 1 sensors-24-03098-t001:** Root File System Packages List.

Function	Package Name
PetaLinux Package Groups	packagegroup-petalinux
	python-modules
	matchbox
	opencv
	self-hosted
	v4lutils
	x11
Apps	autostart
Filesystem packages	Libmali-xlnx
	Opencv
	Libstdc++
	At-spi2-core
	Python3
	Gst-player
	gstreamer
	Gstreamer1.0
	Gstreamer1.0-omx
Modules	dpu
User packages	Dnndk
	libftdi
	Cmake
	Iperf3
	Python-pyserial
	Python3-pip

**Table 2 sensors-24-03098-t002:** CAN protocol for pavement defect detection.

Item	CAN ID	DB0	DB1	DB2	DB3
Obj. Num.	0xE1	Value	-	-	-
DecelerationRamp	0xE2	Type1	distance		-
	0xE3	Type1	-	-	-
Puddle	0xE4	Type2	distance		-
	0xE5	Type2	-	-	-
ExpansionJoint	0xE6	Type3	distance		-
	0xE7	Type3	-	-	-
**DB4**	**DB5**		**DB6**	**DB7**	
-	-		-	-	
Center Coordinates X			Center Coordinates Y		
Length			Width		
Center Coordinates X			Center Coordinates Y		
Length			Width		
Center Coordinates X			Center Coordinates Y		
Length			Width		

**Table 3 sensors-24-03098-t003:** The system information and performance.

	Pavement Defect Detection	
Platform	XCZU7EV	
Model Size	240.59 MB	
Precision	INT8	
DPU Architecture	1xB4096	
DPU Frequency	300 MHz	
Image Size	416 × 416	
Power	8.755 W	
FPS	With video display	Without video display
	~13	~20

**Table 4 sensors-24-03098-t004:** The resource utilization of the FPGA.

Type	Pavement Defect Detection				
Resource	Available	Utilization (Total)	Utilization (%)	Utilization (Resize and UART)	Utilization (%)
LUTs	230,400	66,378	28.8%	133	0.05%
Registers	460,800	6332	1.3%	348	0.07%
BRAM	312	215	68.9%	2	0.6%
DSP	1728	734	42.4%	0	0%

## Data Availability

The original contributions presented in the study are included in the article, further inquiries can be directed to the corresponding authors.

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
