# Peer review of "An Edge Computing System with AMD Xilinx FPGA AI Customer Platform for Advanced Driver Assistance System"

_sensors, 2024, doi:10.3390/s24103098_

Round 1

Reviewer 1 Report

Comments and Suggestions for Authors

Some major comments must be addressed before being considered for publication.

The introduction has provided some useful information for the readers to understand the context of the research. However, the existing research/studies related to the research topic has not been critically reviewed. An in-depth review of the strengths and limitations of existing related studies must be provided in order to justify and explain why the current research topic is worth studying.

The paragraph starts on Line 62 and discusses the research gaps of the study. However, without a clear and systematic review of the existing literature, it is not clear how the research gaps were identified.

The structure of the paper is not well organised. The majority of the paper is about methods/technical details (around ten pages) and the results are less than three pages.

The results section must be improved and more results should be presented.

The discussion section is missing. How the would results/and findings be interpreted?  How the result of the current study stand among the existing literature? How the results/and findings

Again, the conclusion section must be improved. It is currently only one paragraph and less than half a page. What are the limitations of this study? What are the wider implications of the research? What are the future works? 

Comments on the Quality of English Language

Minor editing of English language required

Reviewer 2 Report

Comments and Suggestions for Authors

This paper gives a design based on the AMD Xilinx FPGA customer platform for the ADAS system. The following comments should be considered:

1. There are many other edge computing system such as NXP's embedded system and NVIDIA Jetson AGX Xavier platform, why you using  the AMD Xilinx FPGA customer platform? Are there some special resons or some advantages?

2. Why you using YOLOv3 not using other vesion such as the lastest one?

3. ADAS system has many other fuctions, why you only considering the object detection task?

4. Simulating restluts just give a feasibility verification, what is innovation?

Comments on the Quality of English Language

Null

Reviewer 3 Report

Comments and Suggestions for Authors

This paper presents an innovative edge computing system design specifically tailored for pavement defect detection within the Advanced Driver Assistance Systems (ADAS) domain. The system seamlessly integrates the AMD Xilinx AI platform into a customized circuit configuration, capitalizing on its capabilities. Utilizing cameras as input sensors to capture road scenes, the system employs a Deep Learning Processing Unit (DPU) to execute the YOLOv3 model, enabling the identification of three distinct types of pavement defects with high accuracy and efficiency. This integration of FPGA-based edge computing not only enhances the speed and accuracy of defect detection but also facilitates real-time communication between the vehicle's onboard controller and external systems. This work is meaningful, and I think it is suitable for MDPI Sensors Journal after revision. However, there are still some problems that need to be revised as follows.

1. Multiple images are not explained in the main text of this paper and need to be detailed.

2. Why choose the pavement defect detection function in the ADAS domain for research? Please give elaboration of it.

3. The layout of the three images in Figure 6 needs to be optimized.

4. The red font size in Figure 8 needs to be consistent.

5. As you mentioned, the system can identify three distinct types of pavement defects with high accuracy and efficiency. Why is there only ramp deceleration test in the article? Please give elaboration of it.

Comments on the Quality of English Language

Minor editing of English language required.

Round 2

Reviewer 1 Report

Comments and Suggestions for Authors

Thanks for addressing my comments, the paper has been improved. I have no further comments. 

Comments on the Quality of English Language

English is good enough. 

Author Response

Thank you very much for your insights and valuable feedback, which have greatly contributed to the publication of this manuscript. We sincerely appreciate the time and effort you have invested in reviewing our work. Your comments will undoubtedly enhance the quality and impact of our research. Your guidance is invaluable, and we are grateful for the opportunity to benefit from your expertise.

Reviewer 2 Report

Comments and Suggestions for Authors

It is a revised draft, and the modified contents are highlight. However there are no response documents to reviewers. As pointed in introudtion, the flexibility of FPGA is exploited in this paper. Are there some difficult problem should be solved and how solved should be given out.  

Author Response

Dear Esteemed Reviewer,

We sincerely apologize for any oversight in our previous communication regarding your feedback on our revisions. We appreciate your patience and understanding.

In response to your comments on the initial revisions, we would like to express our gratitude for your valuable insights and suggestions. Your input has been instrumental in enhancing the quality and completeness of our manuscript. We have carefully considered each of your points and have made the necessary revisions accordingly.

Once again, we thank you for your dedication to improving our work and for your continued support throughout this process.

Additionally, the flexibility of FPGA has been thoroughly leveraged in our study, yet maximizing its advantages poses challenges. In the optimization process, a primary concern is balancing performance with resource utilization and power efficiency. To address this, optimized algorithmic designs and high-level synthesis tools can be employed, alongside considerations of practical application costs. Integrating more advanced AI models into FPGA presents opportunities and challenges. Key issues include resource utilization, algorithmic complexity, harnessing parallel processing, enhancing memory bandwidth, power conservation, AI framework integration, and meeting real-time requirements.
